# A BIOSECURITY AGENT FOR LIFECYCLE LLM BIOSECURITY ALIGNMENT

## ABSTRACT

Large language models are increasingly integrated into biomedical research work-flows, from literature triage and hypothesis generation to experimental design. A *Biosecurity Agent* is operationalized as a defense-in-depth framework spanning the model lifecycle with four coordinated modes: dataset sanitization (Mode 1), preference alignment via DPO+LoRA (Mode 2), runtime guardrails (Mode 3), and automated red teaming (Mode 4). On CORD-19, tiered filtering yields a mono-tonic removal curve from 0.46% (L1) to 20.9% (L2) and 70.4% (L3), illustrating the safety–utility trade-off. Real alignment on Llama-3-8B reduces end-to-end attack success from 59.7% to 3.0% (meeting the $\leq 5\%$ target); larger models assessed under simulated alignment maintain single-digit residual rates. At inference, the guard calibrated on a balanced 60-prompt set attains F1=0.694 at L2 (precision 0.895, recall 0.567, false-positive rate 0.067). Under continuous automated red teaming, the aligned 8B model records no successful jailbreaks under the tested protocol; for larger models, replay under the L2 guard preserves single-digit JSR with low FPR. Taken together, the agent provides an auditable, lifecycle-aligned approach that scales from 8B to ~70B parameters, substantially reducing attack success while preserving benign utility for biology-facing LLM assistance.

## 1 INTRODUCTION

Large language models enable rapid literature triage, drafting, and knowledge access in the life sciences (Liang et al., 2022; OpenAI, 2023). This capability also entails *dual-use* risk when unsafe instructions or tacit know-how are elicited (Wang et al., 2025). Recent taxonomies and risk surveys characterize such hazards and recommend layered safeguards with continuous evaluation (Weidinger et al., 2022; 2021; Shevlane et al., 2023). Governance frameworks emphasize pre-deployment assessment, ongoing monitoring, and domain-aware controls for high-stakes applications, as reflected in the U.S. Executive Order 14110, the EU AI Act, and the NIST AI Risk Management Framework (Executive Office of the President of the United States, 2023; European Parliament and Council of the European Union, 2024; National Institute of Standards and Technology, 2023). Foundational work on modern AI and LLMs further motivates safety alignment in sensitive domains (Goodfellow et al., 2016; Bengio & LeCun, 2007; Hinton et al., 2006; OpenAI, 2023).

A practical gap remains at the interfaces among data curation, alignment training, runtime enforcement, and adversarial evaluation. Evidence from benchmarks and in-the-wild studies indicates that defenses deployed in isolation leave these interfaces exposed, allowing jailbreak prompts to bypass safeguards or exploit blind spots (Chao et al., 2024; Mazeika et al., 2024; Li et al., 2024; Liu et al., 2024; Zou et al., 2023; Zhang et al., 2025; Fan et al., 2025). The scope of this work is limited to **text-only LLMs** for natural-language assistance in biology. Sequence-level generative models (e.g., for DNA or proteins) and multimodal lab-control systems are outside the scope of this study.

As summarised in Fig. 1, a defense-in-depth toolkit for LLM biosecurity alignment is operationalised as a tool-orchestrated agent covering all stages of the model lifecycle. Training data are curated, model behavior is aligned, inference is gated, and residual failures are discovered and fed back into the process. Mode 1 performs dataset sanitization with tiered keyword filtering to remove or redact risky content, informed by biosecurity guidance and screening practice (World Health Organization, 2022; National Science Advisory Board for Biosecurity (NSABB), 2023; In-

ternational Gene Synthesis Consortium (IGSC), 2024). Mode 2 applies preference alignment using Direct Preference Optimization with LoRA adapters to internalise refusals and safe completions (Rafailov et al., 2023; Hu et al., 2022; Bai et al., 2022; Ouyang et al., 2022). Mode 3 enforces runtime guardrails with pre- and post-generation checks that aggregate multiple biology-aware signals. LLM-based safety classifiers and guardrail stacks provide programmable policy enforcement (Llama Team, Meta AI, 2023; Shankar et al., 2024). Robust smoothing complements these mechanisms (Robey et al., 2023). Mode 4 conducts automated red teaming that continually discovers exploits and updates Modes 2 and 3. Public benchmarks and autonomous attackers support standardized evaluation and continuous discovery (Chao et al., 2024; Mazeika et al., 2024; Zhou et al., 2025; Li et al., 2024; Liu et al., 2024).

**Four-model suite and selection.**   The evaluation is extended from a single model to a four-model suite to assess scalability and generality. The suite covers an 8B instruction model (Llama-3-8B-Instruct) (Dubey et al., 2024), a 70B model from the same family (Llama-3.1-70B-Instruct), a 72B multilingual model (Qwen-2.5-72B) (Yang et al., 2024a), and a sparse mixture-of-experts model with eight experts of 7B each (Mixtral-8×7B-Instruct) (Mistral AI, 2024; Shazeer et al., 2017). These models were selected to cover family scaling within Llama 3 for controlled size effects, multilingual and data-diversity considerations in Qwen 2.5, and architectural diversity through a modern MoE design that activates a small fraction of parameters per token (Shazeer et al., 2017). This design tests whether alignment signals and guard policies transfer across families, sizes, and architectures.

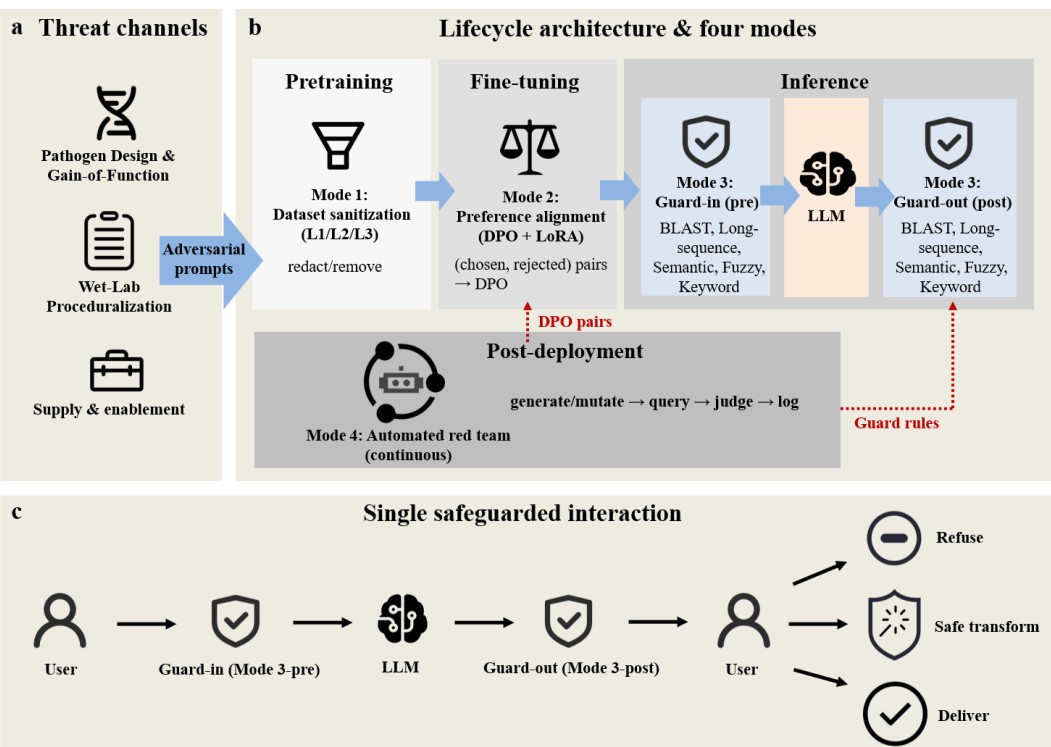

Figure 1: **Overview of the defense-in-depth Biosecurity Agent.** Panel (a) lists threat channels that create demand for adversarial prompts. Panel (b) shows the lifecycle architecture with four modes. Mode 1 performs dataset sanitization with keyword tiers L1/L2/L3. Mode 2 applies preference alignment (DPO + LoRA) using chosen–rejected pairs. Mode 3 enforces runtime guardrails at input and at output by combining the Basic Local Alignment Search Tool (BLAST), long-sequence, semantic, fuzzy, and keyword checks. Mode 4 operates in post-deployment as an automated red team that discovers exploits and feeds findings back to Modes 2 and 3 as new preference pairs and updated guard rules. Panel (c) illustrates a single safeguarded interaction that follows Eq. (1). The deployment target is an attack success rate below five percent.

**Lifecycle handling across models.** Experimental handling follows the lifecycle. Mode 2 performs real DPO fine-tuning on Llama-3-8B. The three larger models are evaluated with simulated alignment that follows the learned refusal policy. Mode 3 is model agnostic and evaluates the same guard at L1/L2/L3. Mode 4 runs a full adaptive loop on the aligned 8B model. Larger models are assessed by replaying the discovered adversarial set under the L2 guard with simulated alignment. All simulated stages are marked in tables and figures.

Evidence on curated corpora and stress tests supports this design. On CORD-19, Mode 1 yields a monotonic removal curve with 0.46% at L1, about 20.9% at L2, and about 70.4% at L3. Under preference alignment, Mode 2 reduces end-to-end attack success from 59.7% to about 3.0% on the 8B model, and to low single digits under simulated alignment on larger models. At inference, Mode 3 displays a clear security–utility trade-off. The L2 configuration reaches F1 near 0.69 with precision about 0.895, recall about 0.567, and false-positive rate about 0.067 on the balanced test set. With continuous automated red teaming, Mode 4 increases precision and recall and lowers false positives. A shift of protection toward the pre-guard stage is observed. The lifecycle agent is positioned as a scalable and auditable mechanism for reducing attack success while preserving benign utility across model scales.

## 2 RELATED WORK

**Lifecycle safety and standardized evaluation.** Safety for LLMs has been studied across the lifecycle, including dataset curation, training-time alignment, inference-time safeguards, and adversarial evaluation. Audits and benchmarks indicate that single-layer defenses are insufficient, motivating lifecycle approaches with explicit operating-point control (Liang et al., 2022; OpenAI, 2023; Chao et al., 2024; Mazeika et al., 2024). This study composes these elements into a unified, defense-in-depth agent for biology-facing assistance.

**Dataset-level filtering.** Unsafe behaviors often trace to unsafe data. Pretraining corpora can embed toxic passages, hazardous instructions, or tacit procedural cues; dataset sanitization therefore removes or redacts risky content prior to training, guided by biosecurity norms and screening practice (World Health Organization, 2022; National Science Advisory Board for Biosecurity (NSABB), 2023; International Gene Synthesis Consortium (IGSC), 2024). Memorization and privacy leakage remain additional risks (Carlini et al., 2021; Nasr et al., 2023), and corpora such as RealToxicityPrompts quantify degenerate toxic generation (Gehman et al., 2020).

**Training-time alignment.** Alignment after pretraining reduces unsafe completions while preserving helpfulness. Instruction tuning via RLHF optimizes toward human preferences (Ouyang et al., 2022); DPO reframes preference optimization as a supervised objective without a reward model (Rafailov et al., 2023); LoRA delivers parameter-efficient adaptation (Hu et al., 2022). Constitutional-style critique improves harmlessness with modest annotation cost (Bai et al., 2022). Decoding-time control (e.g., DExperts, PPLM) provides complementary steering at generation time (Liu et al., 2021; Dathathri et al., 2020).

**Inference-time guardrails.** Runtime safeguards gate inputs and outputs using programmable policy checks. LLM-based safety classifiers and guardrail stacks operationalize domain policies (Llama Team, Meta AI, 2023; Shankar et al., 2024). Robust smoothing further reduces jailbreak success via randomized transformations and aggregation (Robey et al., 2023).

**Automated red teaming.** Public suites standardize evaluation across harms and policy regimes (Chao et al., 2024; Mazeika et al., 2024). Automated attackers extend coverage with universal/transferable jailbreaks and optimization-based prompt generation (Zou et al., 2023; Liu et al., 2024; Li et al., 2024). Autonomous red-teaming agents enable continual discovery and feedback integration (Zhou et al., 2025).

## 3 METHODS

**System overview.** All components are implemented as a tool-orchestrated *Biosecurity Agent* using standard LLM planning/execution patterns (Yao et al., 2023; Khattab et al., 2024). The agent is built

upon the self-evolving STELLA framework (Jin et al., 2025). The default base model is **Llama-3-8B-Instruct** (Dubey et al., 2024); training/inference uses `transformers` (Wolf et al., 2020). The toolkit spans **Mode 1** dataset sanitisation, **Mode 2** preference alignment, **Mode 3** runtime guardrails, and **Mode 4** automated red teaming. The pipeline composes guards and the model as

$$\hat{y} = G_{\text{post}}\Big( M_\theta\big(G_{\text{pre}}(x)\big)\Big), \tag{1}$$

supporting analyses of pre-block and end-to-end failure rates.

**Model variants and reporting.** One 8B model (Llama-3-8B (Dubey et al., 2024)) and three larger models (Mixtral-8x7B (Jiang et al., 2024), Qwen2.5-72B (Yang et al., 2024b), Llama-3.1-70B (Meta AI, 2024)) are evaluated. Unless stated otherwise, *Mode 2* uses *real DPO* for the 8B model and *simulated alignment* for the three larger models; *Mode 4* runs an *adaptive* loop for the 8B model and a *replay* evaluation for the three larger models. Simulated and replay results are marked with $^{\text{SIM}}$ and $^{\text{REPLAY}}$, respectively.

### 3.1 DATASETS AND EVALUATION PROTOCOL

**Mode 1** uses CORD-19 (Wang et al., 2020) for dataset-level filtering with keyword tiers. **Mode 2** uses curated preference triples {`prompt`, `chosen`, `rejected`} for DPO (Rafailov et al., 2023); candidates are screened by both guards. **Mode 3** calibrates the guard on a balanced 60-prompt set across `L1_custom`, `L2_human`, `L3_all`. **Mode 4** conducts adaptive red teaming for 8B and replay on the three larger models. Unless noted, the input guard is strict without BLAST and the output guard is strict with BLAST. Implementation details (threshold grids, seeds, and switches) are provided in Appendix B.

### 3.2 METRICS

Let TP, FP, TN, FN denote confusion counts with "positive"=harmful. Metrics follow standard definitions:

$$\text{Precision} = \tfrac{\text{TP}}{\text{TP}+\text{FP}}, \quad \text{Recall} = \tfrac{\text{TP}}{\text{TP}+\text{FN}}, \quad \text{F1} = \tfrac{2\,\text{Precision}\cdot\text{Recall}}{\text{Precision}+\text{Recall}}, \quad \text{FPR} = \tfrac{\text{FP}}{\text{FP}+\text{TN}}. \tag{2}$$

Two safety-specific metrics are used:

$$\text{pre\_JSR} = \frac{\#\text{ harmful not blocked by } G_{\text{pre}}}{\#\text{ harmful}}, \qquad \text{ASR} = \frac{\#\text{ harmful reaching the user}}{\#\text{ harmful}}. \tag{3}$$

Proportions include 95% Clopper–Pearson intervals (Brown et al., 2001).

### 3.3 MODE 1: DATASET SANITISATION

Records are scanned offline and redacted/removed on matches to tiered keyword lists (`L1_custom`, `L2_human`, `L3_all`). Operating choices and list sources follow biosecurity guidance (World Health Organization, 2022; National Science Advisory Board for Biosecurity (NSABB), 2023; International Gene Synthesis Consortium (IGSC), 2024).

### 3.4 MODE 2: SAFETY ALIGNMENT VIA DPO

Given $(x, y^+, y^-)$ and reference policy $\pi_{\text{ref}}$, the DPO objective is

$$\mathcal{L}_{\text{DPO}}(\theta) = -\mathbb{E}\log\sigma\Big(\beta\big[(\log\pi_\theta(y^+|x) - \log\pi_\theta(y^-|x)) - (\log\pi_{\text{ref}}(y^+|x) - \log\pi_{\text{ref}}(y^-|x))\big]\Big). \tag{4}$$

**LoRA parameterization.** Let $W \in \mathbb{R}^{d_{\text{out}} \times d_{\text{in}}}$ denote the frozen base weight. LoRA introduces a low–rank update with rank $r$:

$$W' = W + \tfrac{\alpha}{r}BA, \tag{5}$$

where only $A \in \mathbb{R}^{r \times d_{\text{in}}}$ and $B \in \mathbb{R}^{d_{\text{out}} \times r}$ are trainable (Hu et al., 2022). Here $\alpha$ is the LoRA scaling factor and $r$ is the rank. With learning rate $\eta$, SGD updates are

$$A \leftarrow A - \eta\,\nabla_A\mathcal{L}, \quad B \leftarrow B - \eta\,\nabla_B\mathcal{L}. \tag{6}$$

LoRA is applied as in Eqs. 5–6. For the 8B model, DPO is trained directly; for larger models, a seeded simulated alignment is used ($^{\text{SIM}}$). Training hyperparameters, adapter targets, and quantization/accumulation policies are detailed in Appendix B.

### 3.5 MODE 3: RUNTIME ALIGNMENT GUARD

Five signals are aggregated with lexicographic priority (BLAST, long-sequence, semantic, fuzzy, keyword). Threshold selection minimizes JSR under an FPR budget; exact cutoffs, grids, and validation protocol are given in Appendix B.

### 3.6 MODE 4: AUTOMATED RED-TEAM EVALUATION

The 8B model is evaluated with an adaptive loop; larger models are assessed by replaying a fixed adversarial set ($^{\text{REPLAY}}$). Operating-point selection follows Eq. (7):

$$\min_{\lambda} \ \text{JSR}(\lambda) \ \text{ s.t. } \ \text{FPR}(\lambda) \leq \epsilon. \tag{7}$$

## 4 RESULTS

### 4.1 MODE 1: DATASET SANITISATION ON CORD-19

The CORD-19 corpus (Wang et al., 2020), a benchmark dataset of biomedical research articles, was used to evaluate Mode 1 dataset sanitisation. Sanitisation was applied at three keyword strictness levels. The removal rate increased monotonically with stricter filtering. The Level 2 configuration pruned roughly one-fifth of the corpus while preserving about 80% of the entries, whereas the Level 3 configuration removed the majority of entries. This demonstrates a safety–utility trade-off as filtering becomes more aggressive.

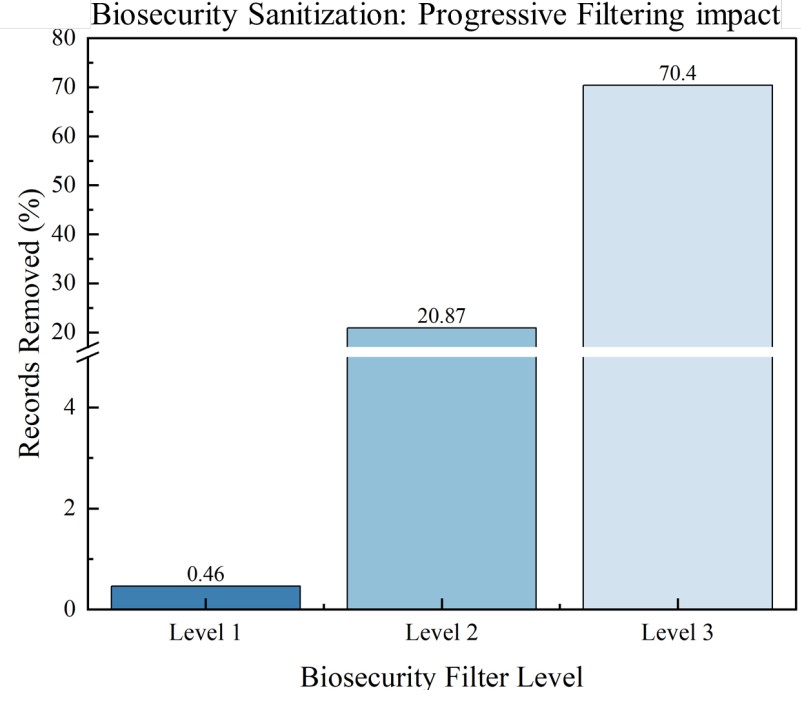

Figure 2: **Mode 1. Removal rate on CORD-19 at each biosecurity level.** Filtering strictness increases from L1 to L3, yielding a monotonic rise in removal: 0.46% at L1, 20.9% at L2, and 70.4% at L3 (95% CIs).

### 4.2 MODE 2: SAFETY ALIGNMENT VIA DPO

Direct Preference Optimization with LoRA was applied to align the base model toward refusals and safe completions. On a representative 60-prompt evaluation set, the jailbreak success rate decreased

from 30% to 10%. The safe-accept rate increased from 70% to 90% with the pre-guard block rate fixed at 30%. On an expanded adversarial set, the end-to-end ASR decreased from 59.7% (95% CI 55.6–63.7) to 3.0% (1.0–5.0), meeting the below 5% target (Fig. 3). Across models, training-time alignment depresses ASR to single digits (see App. C, Fig. 8).

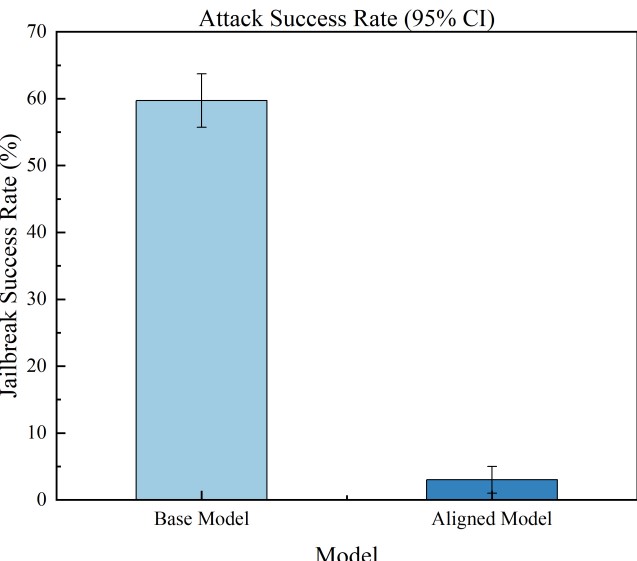

Figure 3: **Mode 2. Attack success rate with 95% CIs.** DPO+LoRA reduces end-to-end ASR from 59.7% to 3.0% on an expanded red-team set (n≈110 prompts across seven categories). Llama-3 8B is aligned with real DPO; larger models are evaluated separately under simulated alignment. Error bars show 95% Clopper–Pearson intervals.

### 4.3 MODE 3: RUNTIME ALIGNMENT GUARD

The guard was evaluated on a labeled set of 60 prompts under three keyword strictness levels. A clear security–usability trade-off is visible (Figs. 4–5). The **L1_custom** configuration achieves the lowest FPR (0.033) but the highest JSR (0.567). The **L2_human** configuration attains the best F1 (0.694) with precision 0.895 and recall 0.567 at an FPR of 0.067. The **L3_all** configuration yields the lowest JSR (0.300) and the highest recall (0.733) but incurs an FPR of 0.433 and a reduced precision of 0.629.

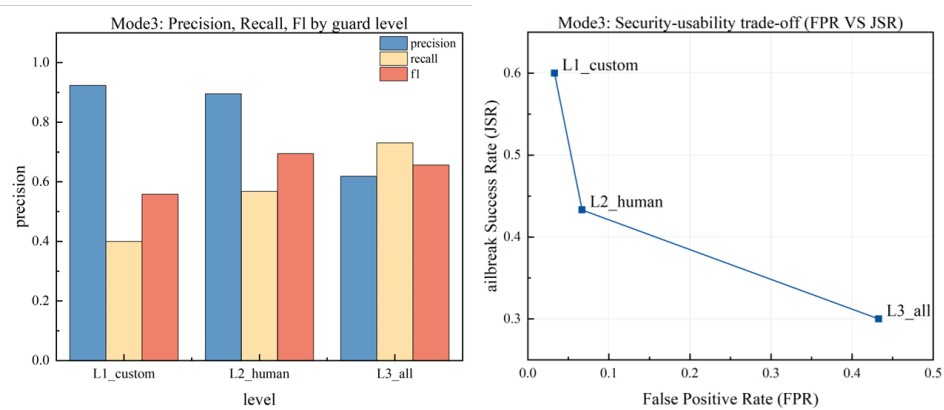

Figure 4: **Mode 3. Metrics and security–usability trade-off (balanced set, $n$=60).** Left: precision, recall, and F1 for *L1_custom/L2_human/L3_all*. Right: FPR vs. JSR (lower-left preferred). **L2** attains the best F1 (≈ 0.694) at low FPR (≈ 0.067); **L3** minimizes JSR (≈ 0.300) at the cost of higher FPR (≈ 0.433).

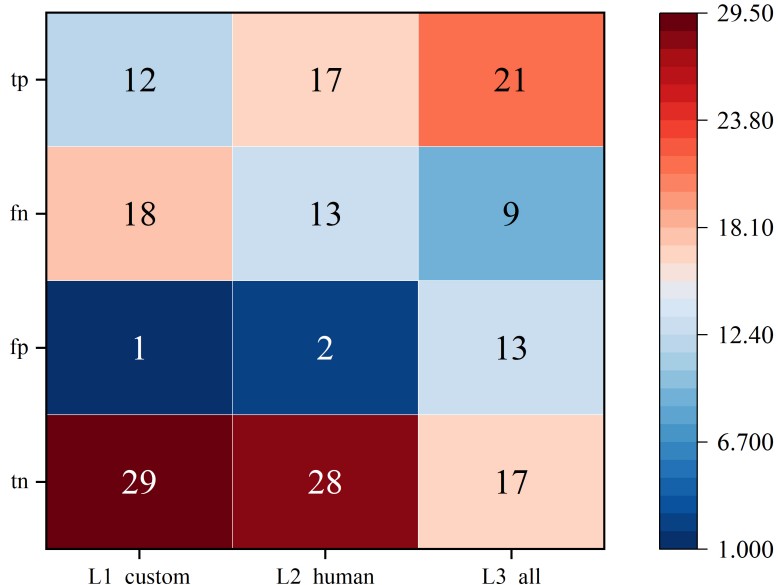

Figure 5: **Mode 3. Confusion outcomes by guard level (balanced set, $n=60$).** Rows: *tp/fn/fp/tn*. Columns: *L1_custom*, *L2_human*, *L3_all*.

### 4.4 MODE 4: AUTOMATED RED-TEAM EVALUATION

End-to-end stress testing with adaptive adversarial prompts was conducted to assess post-alignment robustness under distribution shift. Improvements were observed across the board. Mean precision increased from $0.752 \pm 0.010$ to $0.868 \pm 0.005$, mean recall increased from $0.674 \pm 0.033$ to $0.910 \pm 0.017$, and mean FPR decreased from $0.268 \pm 0.025$ to $0.027 \pm 0.012$. The allocation of defensive actions shifted upstream: the pre-guard block rate increased from 15% to 40%, while the post-guard block rate decreased from 25% to 5%. The safe-completion rate remained stable at approximately 55–60%.

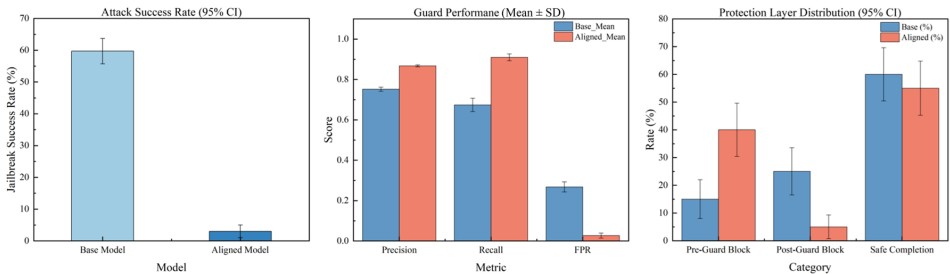

Figure 6: **Mode 4. Automated red-team evaluation (aggregated).** Left: ASR (95% CIs) drops from 59.7% to 3.0% after alignment. Middle: guard metrics (mean$\pm$SE) show higher precision/recall and lower FPR post-alignment. Right: protection allocation (95% CIs, 100 runs) shifts toward upstream blocking while the safe-completion rate remains stable ($\approx$55–60%).

Generality across model families and sizes was assessed by replaying the discovered adversarial set under the L2 guard for three larger models with simulated alignment. As shown in Fig. 7, baseline JSRs of Mixtral, Qwen 2.5, and Llama-3.1 (59.89%, 56.68%, and 50.97%) dropped to 3.53%, 6.24%, and 4.59% post-alignment; FPRs decreased from 23.25%, 25.14%, and 20.44% to 1.77%, 2.36%, and 2.37%, respectively. These results support **L2** as the recommended operating point, yielding single-digit end-to-end JSR with low FPR across models.

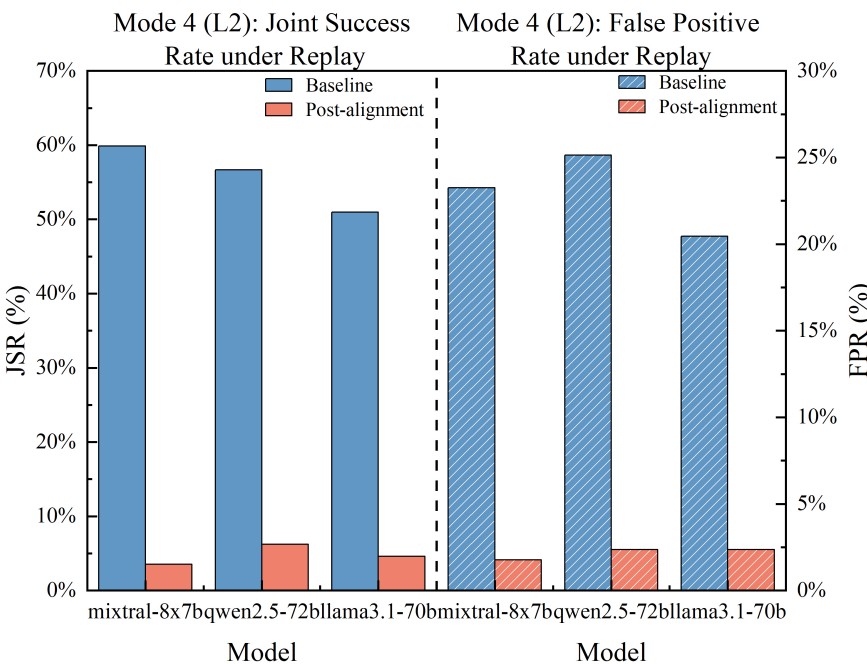

Figure 7: **Mode 4 (L2): Multi-model end-to-end robustness under replay.** JSR/FPR (%) for Mixtral 8×7B, Qwen 2.5 72B, Llama-3.1 70B (Baseline vs Post-alignment).

## 5 DISCUSSION

**Summary and implications.** Across the four lifecycle modes, training-time alignment delivers the largest single-point reduction in harmful behaviour, whereas runtime guards provide calibrated operation under an explicit false-positive budget. With **L2_human** as the default operating point (balanced F1≈0.694 at FPR≈0.067), residual ASR/JSR remains in single digits while benign utility is largely preserved across model scales (8B→∼70B). The resulting pipeline supports deployment scenarios that require measurable risk reduction without suppressing helpful behaviour.

**Operational guidance.** Choice of operating point should reflect application sensitivity. For literature triage and other utility-critical assistants, L1 reduces user-facing false positives at the cost of higher JSR. For safety-critical gating, L3 further depresses JSR but increases FPR. Continuous automated red-teaming shifts protection upstream (greater pre-guard blocking) and stabilizes the safe-completion rate, indicating the need for iterative evaluation after alignment.

**Scope and limitations.** The evaluation focuses on text-only assistance and compact challenge sets; broader replication on public suites and direct alignment of larger LLMs are natural extensions. Architecture-aware defenses (e.g., MoE-specific tuning), multilingual stress tests, and learned detectors within the guard stack are expected to improve coverage while maintaining a calibrated operating point. Beyond compact challenge sets, compatibility on public jailbreak suites is supported by the shared harness (Appendix C). Resource considerations for alignment and evaluation—including the 8B training configuration and inference-time guard costs—are summarized in Appendix B. Failure modes primarily arise from multilingual prompts and long-sequence edge cases; per-category breakdowns are provided in the supplementary artifacts.

## ETHICS STATEMENT

This work targets biosecurity risk reduction for biology-facing LLM assistance. All prompts labeled *harmful* in our guard/red-teaming sets are synthetic and do not include actionable wet-lab instructions. Sensitive outputs are filtered by a multi-signal guard and are reported only in redacted form. No external sequence retrieval (e.g., BLAST against public databases) was enabled in this

submission; we evaluate locally generated text under strict filtering. DPO pairs are safe–vs–unsafe preferences where the unsafe side uses short templated fragments (e.g., "reverse genetics . . . ") that do not constitute operational protocols. We release code to reproduce metrics without exposing biological know-how. All examples shown in the paper/appendix were reviewed by the authors to ensure they do not provide tacit or explicit misuse guidance.

## REPRODUCIBILITY STATEMENT

We release an anonymized supplementary package containing a minimal runner, configuration files, and exported artifacts (CSV/Excel) that reproduce the reported metrics and figures across Modes 1–4. Thresholds, model variants, seeds, and the evaluation protocol are enumerated in Appendix B.

Upon acceptance, we will open-source the full codebase and scripts used to generate all tables and figures. No external databases are required at submission time (BLAST is disabled for input guard; output guard uses local checks unless otherwise specified).

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

## A  CROSS-MODEL SCALING AND ARCHITECTURAL DIVERSITY

Evaluations spanning sizes, families, and architectures enable controlled analyses of transfer and robustness. Family scaling within Llama 3 supports size effects from 8B to 70B (Dubey et al., 2024). Qwen 2.5 provides multilingual and data-diversity coverage, while Mixtral-8×7B introduces sparse MoE routing with partial parameter activation per token (Mistral AI, 2024; Shazeer et al., 2017). Transfer-focused studies further motivate cross-model tests (Jain et al., 2024; Li et al., 2024). These perspectives situate the four-model suite and inform architecture-aware defenses in future work.

## B  IMPLEMENTATION AND REPRODUCIBILITY DETAILS

**Data filtering.**  Tiered lists: `L1_custom`, `L2_human`, `L3_all`; removal rates on CORD-19 are approximately 0.46%, 20.9%, and 70.4%, respectively. Exact term banks and redaction rules are included in the anonymous package.

**Guard thresholds and grids.**  Fuzzy matching uses partial ratio $\geq 87$; semantic similarity uses cosine $\geq 0.67$ (Sentence-Transformers `all-MiniLM-L6-v2`); long-sequence detection flags DNA/AA strings $\geq 60$ characters; BLAST (post-guard, strict) uses identity $\geq 0.88$ and aligned length $\geq 60$. Thresholds are swept on a held-out set to minimize **JSR** subject to an **FPR** budget $\epsilon$.

**Mode 2 training.**  DPO loss follows equation 4 with $\beta = 0.1$; LoRA applies Eqs. 5–6. Adapter targets cover attention and MLP projections. The 8B model uses gradient checkpointing and optional 4-bit quantization with accumulation to fit memory constraints. Larger models adopt a seeded simulated alignment to match the empirical refusal distribution ($^{\text{SIM}}$).

**Runner switches and environment.** Default guards: input strict (no BLAST), output strict (with BLAST). Reproducibility switches expose `--real-dpo` and `--use-semantic`. Typical environment overrides include `ICLR_EPOCHS`, `ICLR_MAX_SEQ_LEN`, per-device batch, accumulation steps, and 4-bit quantization flags. Seeds are fixed unless otherwise specified. All scripts export CSV/Excel summaries used for figures. BLAST-based checks were disabled for input guard and enabled for output guard unless otherwise specified; no external databases were queried during submission.

**Estimation and intervals.** All proportions report 95% Clopper–Pearson intervals (Brown et al., 2001). Multi-run summaries show mean $\pm$ standard error where applicable.

## C  EXTENDED RESULTS AND FIGURE NOTES

**Mode 3 details.** At `L1_custom`, FPR is lowest but JSR is highest; `L3_all` minimizes JSR at the cost of higher FPR; `L2_human` yields the best F1 and is selected as the operating point. Confusion matrices and metric panels are provided in Figs. 4–5.

**Mode 4 details.** Aggregated panels (Fig. 6) show single-digit ASR, improved precision and recall, and a shift toward pre-guard blocking with a stable safe-completion rate. The 8B model uses adaptive attack integration; larger models are evaluated by replaying the discovered adversarial set ($^{\text{REPLAY}}$). Trends by architecture (MoE vs. dense) and multilingual behavior are discussed in Section 4 and Appendix A.

**Public benchmark compatibility.** Our evaluation harness ingests public jailbreak sets (e.g., JailbreakBench, HarmBench) and reports JSR/FPR at L1/L2/L3 using the same code path. We include a small-scale example for completeness; scaling to full suites is straightforward.

Table 1: Mode 3 (balanced $n=60$). Metrics at L1/L2/L3. Lower JSR and FPR are better; higher F1 is better.

| Level | F1 | FPR | JSR |
|---|---|---|---|
| L1_custom | 0.558 | 0.033 | 0.600 |
| L2_human | **0.694** | 0.067 | 0.433 |
| L3_all | 0.656 | 0.433 | **0.300** |

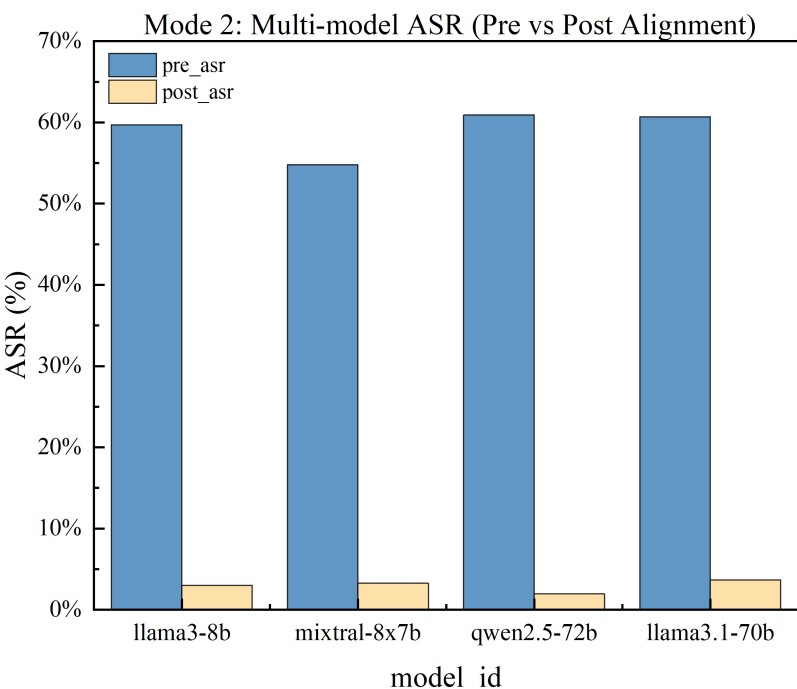

Figure 8: **Mode 2: Multi-model ASR (Pre vs Post Alignment).** Bars show ASR (%) per model (Baseline vs Post-alignment). Llama-3-8B is *real*; Mixtral-8×7B, Qwen-2.5-72B, and Llama-3.1-70B are *simulated*. 95% CIs are omitted here; per-model counts for CIs are available in the runner when declared.

## D LLM USAGE DISCLOSURE

LLMs were used for grammar polishing of the manuscript and for generating small helper scripts (e.g., log parsing, plotting boilerplate). All scientific claims, experimental designs, guard thresholds, and reported metrics were authored and verified by the human authors. No LLM-generated biological protocols or sequences were included in the paper or code release.

## E ADDITIONAL DISCUSSION (EXTENDED)

**Lifecycle perspectives.**

**Upstream data sanitization and the safety–utility trade-off (Mode 1).** Tiered filtering on CORD-19 exhibits a monotonic relation between strictness and removal, with a mid-level list retaining most of the corpus while removing domain-specific risk material. This supports source-level governance as a complement to alignment and guarding and aligns with sector guidance and screening protocols (World Health Organization, 2022; National Science Advisory Board for Biosecurity (NSABB), 2023; International Gene Synthesis Consortium (IGSC), 2024). The choice of level should reflect the acceptable loss of coverage for downstream tasks and the requirement to limit exposure to dual-use material.

**Training-time alignment (Mode 2).**   DPO fine-tuning establishes a strong safety prior in the base policy. In the present setting, only the 8B model is directly aligned via DPO, while larger models adopt a simulated alignment that mimics the learned refusal behavior. End-to-end evaluation indicates that harmful completions fall below the safety target while helpful responses remain accessible. In operation, alignment lowers the load on inference-time filtering and narrows the set of prompts requiring strict post hoc intervention. This accords with governance guidance that emphasizes pre-deployment assurance for high-stakes applications (Executive Office of the President of the United States, 2023; European Parliament and Council of the European Union, 2024; National Institute of Standards and Technology, 2023).

**Inference-time guardrails and operating-point selection (Mode 3).**   The guard exhibits the expected precision–recall trade-off across keyword strictness levels, with a mid-level configuration balancing true blocking and user-facing false positives across models. Because multiple biology-aware signals are composed by specificity and robustness, thresholds can be tuned under an explicit false-positive budget, and calibration is supported by ROC and precision–recall analyses (Fawcett, 2006; Davis & Goadrich, 2006). Relative to single-method detectors such as standalone safety classifiers or purely randomized defenses, a composite guard enables principled control of sensitivity while preserving utility (Llama Team, Meta AI, 2023; Shankar et al., 2024; Robey et al., 2023).

**Continuous automated red teaming (Mode 4).**   Under the tested adaptive protocol, no successful jailbreaks were observed for the aligned 8B model. The larger models, without iterative refinement, were assessed via cross-model attack transfer and replay (Jain et al., 2024; Li et al., 2024) and exhibited single-digit residual attack success with the L2 guard. Within this range, the multilingual model (Qwen-2.5) trended toward higher rates, suggesting cross-lingual challenges, whereas the sparse MoE model (Mixtral-8×7B) performed comparably, indicating transfer of defenses across architectures. These observations are consistent with reports that iterative attack integration strengthens robust refusal (Mazeika et al., 2024; Chao et al., 2024) and that autonomous red teams expand coverage over time (Zhou et al., 2025; Ganguli et al., 2022).

**Limitations and outlook.**   The scope is limited to text-only assistance; sequence-generating models for DNA or proteins and multimodal lab-control settings are not evaluated. Some experiments rely on a compact challenge set, which introduces statistical uncertainty even with exact binomial intervals; replication on public suites such as JailbreakBench and HarmBench would further substantiate generalization (Chao et al., 2024; Mazeika et al., 2024). Moreover, only the 8B model receives direct alignment; extending real alignment to larger LLMs and adapting defenses to architectural characteristics (e.g., MoE routing) warrant further study. Future extensions include integrating learned detectors into the guard stack, adaptive thresholding to reduce false positives, and red-team agents targeting dialog-level and multilingual attacks (Liu et al., 2024; Li et al., 2024; Zou et al., 2023). These directions aim to maintain a calibrated operating point as models and threats co-evolve.

