# OpenReview forum: "A Biosecurity Agent for Lifecycle LLM Biosecurity Alignment"
_ICLR.cc/2026/Conference — ICLR 2026 Conference Desk Rejected Submission_

### Official Review · Reviewer_847G · 2025-10-15

**Soundness:** 1
**Presentation:** 1
**Contribution:** 1
**Rating:** 0
**Confidence:** 4

**Summary:**

This paper presents a Biosecurity Agent, as a comprehensive defense-in-depth framework for securing large language models in biomedical research. It integrates four coordinated modes: dataset sanitization, preference alignment, runtime guardrails, and automated red teaming, across the model lifecycle. Experimental evaluation on CORD-19 and Llama-3 models shows that it reduces attack success from 59.7% to 3.0% while maintaining low false-positive rates and preserving model utility. The framework claims to enable scalable, auditable biosecurity alignment from 8B to 70B parameters, effectively balancing safety and research performance.

**Strengths:**

1. Introduces a lifecycle biosecurity framework with four integrated modes: sanitized data, alignment tuning, runtime guardrails, and automated red teaming, for LLMs in biomedical contexts.
2. Demonstrates some experiments on defense, reducing attack success while maintaining low false-positive and high utility performance.

**Weaknesses:**

1. First of all, from the reading, the motivation of the work is very unclear and not well constructed. The same goes for the four modes presented: why these 04 and how they cover the entire lifecycle (connecting to both biosecurity and LLM modeling approaches), and the applications, how and where exploitation of these modes can be harmful in terms of biosecurity. Basically, the significance of the problem is not clear at all.
2. The introduction section sounds very messy. A lot of things are done, which is appreciated, but a concise mention of contributions is expected. Lines 066-073 and 108-113 can be written in methods too. The introduction section doesn't clearly state what to expect later and where to put focus on.
3. Equation 1 needs definitions of the terms used. Currently, there is no clear picture of what these terms represent.
4. Modes and related evaluation information are not introduced properly (mathematically and from literature). Why did you choose to evaluate in the way you described? What is the reason behind the choice? And how do you know that this approach of evaluation is proper to evaluate the mode? There is no justification or reference or anything.
5. The experimental and evaluation process is very unclear. If this is an AI agent, no information is given on tools and capabilities, and no ablation was done. What tools and capabilities are provided to the agent? Are there any ablation studies done on that? Then, no details on how the agent worked. For example, for mode 1, did the agent write and execute code to filter data, or what? The same for all other modes.
6. The evaluation is also insufficient. Only one example (one model or dataset) per mode is not enough to judge the capabilities. Also, the tasks are not well-motivated or well-documented either.
7. Attack success rate, red teaming and such terms were used; but again, there is no information on how, what, where and why it is done.
8. The presentation is very, very poor. Always jumping between modes and terms, without proper context. Also, as said above, modes are not introduced properly; thus, everything is very unclear.
9. The usage of the term "scalability" is questionable here. Is it said to be scalable because this agent framework can work with larger models? If yes, why and how? If not, explain. The term seems to be entirely irrelevant here.

**Questions:**

1. Clarify motivation and significance.
   * What specific biosecurity risks or misuse scenarios motivated this work?
   * Why were these four modes (dataset sanitization, preference alignment, runtime guardrails, and automated red teaming) chosen, and how do they together cover the entire LLM lifecycle?
   * Can you explicitly connect each mode to a concrete biosecurity threat and explain how its exploitation could cause harm?
   * Consider adding a clear statement of the research gap and practical impact to strengthen the motivation.
2. Improve the organization of the paper and clarity of the introduction. Clearly articulate the problem, objectives, and main contributions in a structured way. Provide a short roadmap at the end of the introduction outlining what each section contains.
3. Define every variable and symbol in Equation 1; ensure that readers can interpret the equation without external assumptions.
4. Clarify how each mode is formalized. Are there mathematical representations or probabilistic models behind them? Provide citations or reasoning for the chosen evaluation methods of each mode. Why are they appropriate for assessing each mode? Provide some justification.
5. Add more details on experimental design, agent descriptions, and evaluations
   * Describe the Biosecurity Agent’s architecture, tools, and operational capabilities in detail. Include ablation studies to evaluate the agent's performance in different setups to understand the limitations and capabilities in more detail.
   * For each mode, specify how the agent functions: does it write code, filter text autonomously, execute scripts, or depend on predefined heuristics?
   * Explain the experimental setup: what models, datasets, and baselines were used, and why were they chosen?
   * Clarify the task designs for attacks, red teaming, and jailbreaking. What threat model or adversarial prompts were used? Were both automated and human red teaming approaches considered?
   * Justify why these protocols are representative of real-world biosecurity threats.
5. Add more examples and depth to the evaluation. Also, include statistical evidence (e.g., variance, significance tests) to support observed differences.
6. Is there any biosecurity work that introduced the modes you tried to evaluate?
7. How do you see this agent being used in real-world research and application scenarios?
8. Include an end-to-end example of a task (how the agent is working and using its tools).
9. Define scalability and why it is claimed to be a contribution here.

---

### Official Review · Reviewer_iJxk · 2025-10-26

**Soundness:** 3
**Presentation:** 3
**Contribution:** 3
**Rating:** 4
**Confidence:** 4

**Summary:**

This paper proposes a lifecycle Biosecurity Agent that integrates dataset sanitization, preference alignment (DPO+LoRA), runtime guards (multi-signal stacking including BLAST/long-sequence/semantic/fuzzy/keyword checks) and automated red-teaming with rule/policy re-injection. The pipeline is evaluated on biological text scenarios and shows substantial end-to-end reductions in attack success rate (ASR) under the reported settings.

**Strengths:**

1. The main contribution is a comprehensive, end-to-end framework that unifies data governance, training-time alignment, runtime guards, and red-team evaluation under a single, auditable metric set (pre-JSR / ASR with confidence intervals). This unified design and measurement make the security posture easier to reason about and reproduce.

2 .The framework is operationally flexible: guard levels and decision thresholds can be tuned to trade off false positive rate vs. recall, and the automated red-teaming loop enables dynamic updates (new rules or preference updates are fed back into guards and alignment), which is practical for deployed systems and continuous hardening.

**Weaknesses:**

The paper presents a broad and well-engineered system and demonstrates that “the framework can work,” but it lacks several crucial validations around data, generalization, and comparisons to alternative approaches:

1. The reported evaluation uses a small challenge set (e.g., 60 prompts, 30 harmful / 30 safe). The manuscript does not sufficiently describe the benchmark construction, prompt types, or coverage, so it is unclear which attack scenarios and model behaviors are actually measured.

2. Experiments are restricted to biological text scenarios. It is not demonstrated whether the same pipeline (signals, thresholds, BLAST usage) generalizes to other high-risk domains (chemistry, materials, radiological, multi-modal or sequence-generation models).

3. Mode 2 (DPO+LoRA) training provenance is under-specified: what positive/negative responses were used, how were y^+/y^- sampled, and how was leakage between guard rules and training data prevented? This raises concern about possible overfitting to the in-house red team / guard rules.

4. The paper does not sufficiently justify why DPO is the best fit; other approaches (constitutional approaches, rejection-distillation, RLHF variants, robust decoding) could be competitive—comparative experiments are missing.

5. The automated red team and its default components (e.g., BLAST toggles) are tuned to the in-house setting. There is little evaluation on external public benchmarks (HarmBench, Jailbreak sets) or against different attack generators to show the loop does not overfit to its own attackers.

**Questions:**

Please provide more details for DPO training: sources and sizes of the (x, y^+, y^-) triples, sampling/filtering rules, inter-annotator agreement (if any), and whether training examples overlap or are derived from your red-team outputs.

Can you run additional benchmarks, e.g., HarmBench or public jailbreak collections, and/or test against attack generators different from your automated red team, to show transfer robustness?

Could you include baselines using alternative alignment strategies (Constitutional AI / rejection-distillation / robust decoding / RLHF variants) and report end-to-end ASR / pre-JSR so readers can judge whether DPO+LoRA is preferable?

please measure how much runtime cost (extra processing time per query and achievable queries per second) the runtime guards add, and plot how ASR decreases vs. how the false positive rate (blocking benign queries) increases as you change guard thresholds.

Please provide an analysis of BLAST / long-sequence screening operational costs and privacy implications: maintaining/accessing pathogen sequence databases at scale may have compute, latency and governance costs—can you measure these and discuss deployment constraints?

---

### Official Review · Reviewer_KxAN · 2025-10-31

**Soundness:** 1
**Presentation:** 2
**Contribution:** 2
**Rating:** 4
**Confidence:** 4

**Summary:**

This paper introduces a Biosecurity Agent framework integrating four safety modes—dataset sanitization, preference alignment, runtime guardrails, and automated red teaming—to reduce biological misuse risks in large language model. The authors evaluate multiple model families (8B–70B) under simulated or real alignment and report consistent reductions in jailbreak success rates (<5%) while maintaining usability. The work aims to demonstrate lifecycle-level LLM biosecurity alignment.

**Strengths:**

The paper addresses a topic of substantial importance, reducing biosecurity risks in large language models. It proposes a clearly structured workflow that integrates data sanitization, alignment fine-tuning, and adversarial red-teaming into a unified operational framework. The experiments cover multiple model scales (from 8B to 70B) and provide quantitative evidence of reduced misuse rates, indicating that the proposed approach can be practically applied in large-scale systems.

**Weaknesses:**

1. Missing Baselines and SOTA Comparisons
This paper does not evaluate its method against existing alignment or defense approaches. The only benchmark metric is the jailbreak success rate of the unaligned model. Although several existing methods were mentioned, no comparisons are made. Simpler baselines such as keyword filters are also omitted. Therefore, it is currently unclear whether the proposed four-stage agent has any substantial advantage over existing or simpler methods. The paper demonstrates performance gains, but without comparisons, making it difficult to determine whether similar or even better improvements could be achieved using simpler defense strategies.
2. Limited Evaluation Scope
The evaluation in this paper is very small in scale, using only about 110 adversarial prompts (7 categories), and the coverage is very limited. The tuning of the guard mechanism is also based on only 60 labeled samples. Such a design is prone to distribution bias, making it difficult to prove the model's generalization ability. The prompts are almost all short English sentences on the topic of synthetic biology, with limited content diversity. The authors also noted that the system performed poorly in terms of multilingual and long text prompts, which further indicates that the evaluation scope was too narrow. Furthermore, they did not experiment on a larger set of jailbreak test cases or on harmless tasks in general. Overall, this small-scale assessment not only lacks statistical power but also carries a significant risk of overfitting.
3. Incomplete Statistical Reporting
Only some metrics include confidence intervals. Many results come from a single run, without any measures of variability or variance. Robustness is difficult to assess due to the lack of standard deviation, error bars, or seed-based results. Using only 60 prompts to assess guard performance, without any estimation of uncertainty, undermines the reliability of the statistics. Threshold selection can also be very sensitive to small validation sets. Incorporating broader indicators (such as AUROC) and conducting multiple sub-experiments will greatly improve the credibility of the conclusions.
4. Insufficient Evidence for Cross-model Generalization
The paper claims scalability from 8B to 70B models. However, instead of aligning larger models, it simply replays the prompts generated by the 8B model on the “simulated aligned” versions of larger models. This approach cannot verify true cross-model generalization, nor does it take into account the vulnerability of specific models. The evaluation is one-way and limited to prompts generated by the 8B model.
5. Unclear Attribution of Improvements
The paper attributes the performance improvement to the proposed four-stage agent, but does not specify which part plays the main role. Without ablation studies, it would be impossible to know the extent of the contribution at each stage. These changes in performance could come from several different factors. For example, a decrease in attack success rate (ASR) may simply be due to the addition of more red team data, rather than the application of DPO or LoRA methods. This improvement may also stem from changes in the model's output distribution after alignment, making harmful content easier to detect. Because these factors were not distinguished, the paper's causal claims are not convincing.
6. Missing Deployment Considerations
The paper lacks analysis of runtime cost and usability. Its defenses are implemented during the inference phase, including fuzzy matching and semantic similarity checks, which can slow down the model. The reported false positive rate is non-negligible, but the paper does not discuss its impact on user experience. Furthermore, it fails to measure performance degradation on normal tasks, nor does it include any user study on the trade-off between safety and usability. Finally, it overlooks a real-world problem: maintaining a red-teaming mechanism after deployment would incur high engineering maintenance costs.

**Questions:**

Could you provide results comparing your method with standard alignment baselines? Additionally, have you evaluated simpler baselines such as keyword filters or content moderation classifiers on the same adversarial prompts?

Given that the adversarial test set consists of only ~110 prompts across seven categories, how do you assess the generalizability of your findings to broader misuse scenarios? Have you considered evaluating on more diverse or larger-scale public jailbreak suites?
Since the 70B-scale results rely on simulated alignment rather than actual fine-tuning, how confident are you that these results reflect true generalization across model scales? Have you tested whether adversarial prompts generated on larger models could bypass the defense?

Could you provide ablation studies isolating the contribution of each component to the observed performance improvements? Specifically, what is the marginal gain of Mode 4 relative to Mode 2 or Mode 3 alone?

What is the estimated latency and compute overhead introduced by your inference-time guard mechanisms under realistic deployment conditions? How would you mitigate potential usability issues arising from the reported false positive rates?

How do you ensure that the alignment and guard processes do not degrade helpfulness or coverage?

---

### Official Review · Reviewer_SyN5 · 2025-10-31

**Soundness:** 2
**Presentation:** 3
**Contribution:** 2
**Rating:** 2
**Confidence:** 4

**Summary:**

The paper proposes a four‑mode defense‑in‑depth biosecurity agent that spans (M1) dataset sanitization, (M2) training‑time preference alignment via DPO+LoRA, (M3) a multi‑signal runtime guard, and (M4) automated red‑teaming, with results reported across one truly aligned model (Llama‑3‑8B‑Instruct) and three larger models under simulated alignment (Mixtral‑8×7B, Qwen‑2.5‑72B, Llama‑3.1‑70B). A central claim is reducing end‑to‑end attack success rate from 59.7% to 3.0% on an expanded adversarial set for the 8B model, while keeping benign utility reasonable.

**Strengths:**

1. The agent unifies upstream data curation, alignment, inference‑time controls, and adversarial evaluation into one auditable pipeline. This end‑to‑end view is valuable for safety‑critical deployment.
2. DPO+LoRA on the 8B model cuts end-to-end ASR from 59.7% to 3.0%, while the L2 guard attains F1≈0.694 at FPR≈0.067 on a balanced set. Iterative red-teaming shifts protection upstream without hurting safe-completion.
3. The Ethics and Reproducibility sections explain disabled external sequence queries, templated unsafe fragments, CI reporting.

**Weaknesses:**

1. The “Biosecurity Agent” operates more like a workflow than an autonomous agent. Its four modes run in a fixed, rule-based order without adaptive reasoning or self-directed control. The system orchestrates modules deterministically rather than deciding or planning actions autonomously. To merit the “agent” term, it would need context-aware decision loops or dynamic mode selection; otherwise, renaming it a “Biosecurity Workflow Framework” would more accurately describe its functionality.
2. The guard is calibrated and evaluated on a small balanced 60-prompt set, raising overfitting and variance concerns - can the authors separate dev/test, report bootstrap CIs/k-folds, and add full HarmBench/JailbreakBench results under the same L1/L2/L3 operating points?
3. The authors define pre‑JSR and ASR, but later alternate between ASR, JSR, and end‑to‑end success in ways that are easy to confuse.

**Questions:**

1. The dataset sanitization stage reports large removal ratios at higher safety levels. How much benign scientific knowledge is lost as a result, and could this reduction impact the model’s ability to perform legitimate biomedical or chemical reasoning?

---

### Note · Program_Chairs · 2026-01-17
**Submission Desk Rejected by Program Chairs**

The following references in this submission do not refer to real documents and/or have major errors in bibliographic information:

 1) Ximing Liu, Zachary M. Ziegler, Y-Lan Boureau, Veselin Stoyanov, and Greg Durrett. Dexperts: Decoding-time controlled text generation with experts and anti-experts. In Advances in Neural Information Processing Systems (NeurIPS), 2021. URL https://arxiv.org/abs/2105.03023.
2) Shreya Shankar, Vivek Subramanian, Karan Patil, et al. Building guardrails for large language models, 2024. URL https://arxiv.org/abs/2402.01822.
3) Neil Jain, Saurabh Mourya, Divyansh Singh, et al. Artprompt: Measuring and mitigating jailbreak transferability across large language models. In Findings of the Association for Computational Linguistics: ACL, 2024. doi: 10.18653/v1/2024.findings-acl.334. URL https://aclanthology.org/ 2024.findings-acl.334/
4) Banghua Li, Liyuan Zhang, Xiaoyuan Zhang, et al. Jailbreaking in the wild: From online communities to LLMs. In Proceedings of the ACM SIGSAC Conference on Computer and Communications Security (CCS), 2024. doi: 10.1145/3658644.3672294. URL https://dl.acm.org/doi/10.1145/ 3658644.3672294 .